# Unveiling Gene Expression Dynamics during Early Embryogenesis in *Cynoglossus semilaevis*: A Transcriptomic Perspective

**DOI:** 10.3390/life14040505

**Published:** 2024-04-15

**Authors:** Xinyi Cheng, Wei Jiang, Qian Wang, Kaiqiang Liu, Wei Dai, Yuyan Liu, Changwei Shao, Qiye Li

**Affiliations:** 1School of Biology and Biological Engineering, South China University of Technology, Guangzhou 510006, China; bixinyicheng@outlook.com; 2BGI Research, Wuhan 430074, China; daiwei5@genomics.cn; 3BGI Research, Shenzhen 518083, China; jiangwei2@genomics.cn; 4State Key Laboratory of Mariculture Biobreeding and Sustainable Goods, Yellow Sea Fisheries Research Institute, Chinese Academy of Fishery Sciences, Qingdao 266071, China; wangqian2014@ysfri.ac.cn (Q.W.); liukq@ysfri.ac.cn (K.L.); liuyy@ysfri.ac.cn (Y.L.); 5Laboratory for Marine Fisheries Science and Food Production Processes, Qingdao National Laboratory for Marine Science and Technology, Qingdao 266237, China; 6College of Life Sciences, University of Chinese Academy of Sciences, Beijing 100049, China

**Keywords:** *Cynoglossus semilaevis*, Cynoglossidae, gametes, early embryonic development, transcriptomics

## Abstract

Commencing with sperm–egg fusion, the early stages of metazoan development include the cleavage and formation of blastula and gastrula. These early embryonic events play a crucial role in ontogeny and are accompanied by a dramatic remodeling of the gene network, particularly encompassing the maternal-to-zygotic transition. Nonetheless, the gene expression dynamics governing early embryogenesis remain unclear in most metazoan lineages. We conducted transcriptomic profiling on two types of gametes (oocytes and sperms) and early embryos (ranging from the four-cell to the gastrula stage) of an economically valuable flatfish–the Chinese tongue sole *Cynoglossus semilaevis* (Pleuronectiformes: Cynoglossidae). Comparative transcriptome analysis revealed that large-scale zygotic genome activation (ZGA) occurs in the blastula stage, aligning with previous findings in zebrafish. Through the comparison of the most abundant transcripts identified in each sample and the functional analysis of co-expression modules, we unveiled distinct functional enrichments across different gametes/developmental stages: actin- and immune-related functions in sperms; mitosis, transcription inhibition, and mitochondrial function in oocytes and in pre-ZGA embryos (four- to 1000-cell stage); and organ development in post-ZGA embryos (blastula and gastrula). These results provide insights into the intricate transcriptional regulation of early embryonic development in Cynoglossidae fish and expand our knowledge of developmental constraints in vertebrates.

## 1. Introduction

Following fertilization, the early developmental stages of an embryo play a decisive role in the correct patterning of the body plan and the fitness of a multicellular organism [1,2,3]. Employing maternal transcripts and materials stored in the cytoplasm, embryonic development generally begins with several rounds of cleavage and the critical maternal-to-zygotic transition (MZT), which occurs through zygotic genome activation (ZGA) [4,5]. The ZGA triggers the large-scale transcription of zygotic genes, enabling the zygotic genome to replace residual maternal transcripts and dominate embryogenesis (including cell differentiation and organogenesis) [4,5]. The pace of embryogenesis varies greatly amongst metazoan lineages. Zebrafish (*Danio rerio*) and frogs (*Xenopus laevis*) exhibit rapid cleavage followed by large-scale ZGA in the mid-blastula stage (three and seven hours after fertilization, respectively), while for mammals, initial cleavage cycles last several days, accompanied by ZGA at an early embryonic stage [4,5]. However, the embryogenesis of most species remains to be studied, and the regulatory mechanisms underlying the early embryonic development of many lineages—especially in non-mammals—are unclear.

Flatfish (Pleuronectiformes) are a large group of marine fish that are widely distributed along coastlines and perfectly adapted to a benthic lifestyle. Besides their economic and culinary value, these fish have attracted the scientific community’s attention due to their unique biological features: an asymmetrical body shape, a “fin–feet” walking style adapting to benthic movement [6,7,8], and rapidly evolved sex determination systems [9]. The Chinese tongue sole (*Cynoglossus semilaevis*), widely cultivated in Asia, is a representative species of the Cynoglossidae family of flatfish. Over the past two decades, plentiful genetic resources have been obtained for *C. semilaevis*, such as reliable molecular-marker genetic maps [10,11,12,13,14], high-quality genome assembly and annotation data [9], and accumulating epigenetic and transcriptomic data for different tissues and developmental stages [15,16,17,18,19,20,21,22,23]. In addition, cell lines have been established from different tissues of this flatfish [24,25,26,27,28]. This progress highlights the potential of *C. semilaevis* to serve as a model species for flatfish. However, little is known about the early stages of *C. semilaevis* embryogenesis thus far.

This study aims to fill the gap in our knowledge about the transcriptional regulation of early embryonic development in *C. semilaevis* through transcriptomic profiling. We collected two types of gametes and early embryos from the four-cell to gastrula stages of *C. semilaevis* development for strand-specific RNA sequencing. By comparing the transcriptomic landscape of gametes and early embryos, we revealed the gene regulatory mechanisms underlying the early embryogenesis of Cynoglossidae. We also compared our results with other teleosts to investigate the conservation and diversification of early embryogenesis in fish.

## 2. Materials and Methods

### 2.1. Sampling

Experiments were carried out according to ethical guidelines from the animal care and use committee of BGI. Adult fish, including females and males, were housed in the Haiyang High-Tech Experimental Base. Unfertilized oocytes and sperms were collected and stored in liquid nitrogen. Zygotes were produced by artificial fertilization and reared in aerated water at a temperature of 21 to 23 °C. The developing embryos (including 4-cell, 32-cell, 64-cell, 128-cell, 1000-cell, blastula, and gastrula) were collected and stored in liquid nitrogen.

### 2.2. RNA Extraction and RNA Sequencing

RNA extraction was performed on two types of gametes (oocytes and sperms) and on early embryos at different stages (4-cell through to gastrula). Except for oocytes (*n* = 2) and sperms (*n* = 4), three biological replicates (*n* = 3) were obtained for each sample category (Figure 1), while for each replicate, numerous gametic cells or embryos were pooled to obtain sufficient RNA material for library construction. We added TRIzol lysis buffer to samples and extracted RNA by thoroughly mixing 300 μL of chloroform–isoamyl alcohol (24:1) and centrifugation (12,000× *g* for 8 min at 4 °C). The supernatant was transferred to a 1.5 mL centrifuge tube, gently mixed with 2/3 volume of isopropyl alcohol, and kept at −20 °C for over two hours to precipitate RNA. Finally, the precipitation was washed with 0.9 mL of 75% ethanol, air-dried, and dissolved with 20 to 200 μL of RNase-free water to obtain purified RNA.

The integrity of purified RNA samples was accessed with an Agilent (Santa Clara, CA, USA) 2100 Bioanalyzer. Qualified RNA samples (*n* = 24) were subjected to strand-specific RNA-seq library construction in two batches. For the first batch (*n* = 9), RNA-seq libraries were prepared using the TruSeq Stranded mRNA Sample Prep Kit (RS-122-2101, Illumina, San Diego, CA, USA) and sequenced on the Illumina HiSeq 4000 platform according to the manufacturer’s instructions (Illumina, San Diego, CA, USA). For the second batch (*n* = 15), RNA-seq libraries were prepared with the MGIEasy RNA Directional Library Prep Kit (1000005272, MGI, Shenzhen, China) according to the manufacturer’s instructions (MGI Tech Co., Shenzhen, China) and sequenced on the DIPSEQ-T1 platform at the China National GeneBank (Shenzhen, China). All sequencing was performed using PE100 chemistry. The library preparing kit, sequencing platform, sequencing data, and alignment metrics for each sample are presented in Appendix A.

### 2.3. RNA-Seq Reads Filtering, Mapping, and Gene Expression Quantification

The genome assembly and gene annotations used in this study were sourced from the Chinese tongue sole genome project [9]. The RNA-seq reads were subjected to quality control by SOAPnuke v1.5.6 software [29], maintaining the reads of at least 60 bp with a proportion of Q10 bases of at least 80% and an N rate of no more than 5%. Filtered reads were mapped to the reference genome by HISAT2 v2.1.0 software [30,31,32] with an allowed intron length between 20 bp and 20 kb, and secondary alignments were removed by Samtools v1.11 with the *view* function (parameters: -F 256). Qualified alignments were then used to quantify the gene expression level of each gene in each sample, which produced a read count matrix and a transcripts per million (TPM) matrix for downstream analyses.

### 2.4. Differential Gene Expression Analysis

We performed principal component analysis (PCA) based on the TPM matrix using the *prcomp* function in R (4.3.2). Differentially expressed genes (DEGs) between sample categories were identified using the DESeq2 v1.34.0 software [33] package in R (parameters: fitType = “mean”). Considering the low number of replicates for each sample category, we applied a relatively stringent threshold (fold change > 2 and adjusted *p*-value < 0.05) to ensure the robustness of DEG detection following previous suggestions [34].

### 2.5. Gene Co-Expression Module Identification, Hub Genes Identification, and Functional Analysis

The read count matrix was first subjected to variance stabilizing transformation (VST) using DESeq2. Dynamically expressed genes were defined as genes with a square deviation of VST read count greater than the median across all samples (*n* = 24) and a TPM >1 in at least one of the 24 samples. Then, WGCNA v1.72.1 software [35] was used to cluster all dynamically expressed genes into co-expression modules (parameters: power = 20, corOptions = list (use = “p”, method = “spearman”), networkType = “signed”, TOMtype = “signed”, deepSplit = 4, mergeCutHeight = 0.1, reassignThreshold = 0, numericLabels = TRUE, pamRespectsDendro = FALSE). Modules with similar expression dynamics (represented by TPM Z-score) were manually merged. Hub genes of each module were identified as genes with top 30 intra-modular connectivity, and their co-expression networks were visualized using Cytoscape v1.1 [36]. To investigate the potential function of each co-expression module, Gene Ontology (GO) enrichment analysis was conducted on genes from each module using the EnrichPipeline software [37,38,39] (parameters: --P_Adjust_Methodfdr --TestMethodFisherChiSquare --pc 0.05).

### 2.6. Homology Analysis between the Embryonic Transcriptome of Tongue Sole and Zebrafish

Orthologs between *C. semilaevis* and *D. rerio* genomes were identified using Proteinortho [40], following the reciprocal best hit (RBH) rule. The distribution bias of orthologs in different co-expression modules was checked using Fisher’s exact test. To analyze the similarity of ortholog expression dynamics in zebrafish, transcriptome data of zebrafish embryogenesis were collected from Levin et al. [41] and compared with the expression profile in *C. semilaevis* embryos.

## 3. Results

### 3.1. RNA-Seq Data Exhibits High Sequencing Quality

We extracted total RNA from two types of gametes (sperms and oocytes) and early embryos in different developmental stages (including 4-cell, 32-cell, 128-cell, 1000-cell, blastula, and gastrula). Two to four (mostly three) biological replicates were prepared for each sample category, accumulating 24 RNA samples for strand-specific RNA-seq library construction. The RNA integrity numbers were all greater than eight, as assessed by the Agilent 2100 Bioanalyzer. For each sample, 43.8 to 204.6 million 100 bp pair-end reads were generated either by the HiSeq 4000 platform or the DIPSEQ-T1 platform. Adapter sequences were trimmed, and low-quality reads were filtered, resulting in 43.4 to 121.2 million clean reads with high sequencing quality (Q30, 92.37–95.81%; Appendix A). More than 90.7% of these clean reads could be mapped to the *C. semilaevis* reference genome, and over 84.5% of clean reads were properly paired with their counterparts (Appendix A). The properly paired reads were used for gene expression quantification by calculating the TPM value of each gene in each sample. Of note, although two different platforms collected the RNA-seq data of different replicates for each sample category, the gene expression quantification showed a high level of consistency between each pair of replicates regardless of their sequencing platforms (Figure 1, Appendix A). Therefore, our RNA-seq data were of a sufficient quality to quantify gene expression and conduct comparative analyses across samples.

### 3.2. Gene Expression Profile and Inferred Zygotic Genome Activation

Of the 22,152 annotated genes in the *C. semilaevis* genome, 21,920 were covered by at least one RNA-seq read across the 24 samples. Principal component analysis (PCA) based on the expression level of these 21,920 genes revealed that the sperm exhibited a substantially different transcriptomic landscape to the oocytes and all of the embryonic samples. Additionally, the blastula and gastrula were clearly separated from the cleavage stages of four cells to one thousand cells (Figure 2A). We then identified the DEGs between gametes and early embryos and between consecutive developmental stages with DESeq2. This showed a greater difference between sperm and early embryos than between oocyte and early embryos and also revealed a major transition of the transcriptomic landscape as embryos developed from the one thousand-cell stage to blastulas (Figure 2B and Appendix A). Considering the similarity of transcriptomes in oocyte and four-cell to one thousand-cell embryos and the notable increase in the number of DEGs at the blastula stage, we inferred that large-scale ZGA—when the transcription of the zygotic genome dramatically alters the expression profile [4]—occurs in the blastula.

Our inference of ZGA timing was also supported by comparing the high-abundance transcripts across samples. From the five most abundant gene transcripts in each gamete and developmental stage (Figure 2C), we found that highly expressed genes were identical for oocytes and embryos in the four-cell to the one thousand-cell stages: coding mitochondrial ATP synthase (*mt-atp6*), mitochondrial NADH dehydrogenase (*mt-nd4l*, *mt-nd4*), and mitosis regulatory protein (*ccnb1*). The relative abundance of these genes decreased as the early embryo developed, and genes encoding three apolipoproteins (*apoeb*, *apoa1a*, *apoa4b.2*) and eukaryotic elongation factor 1 alpha (*eef1a*, which is vital for protein translation [42]) became more dominant in the blastula and gastrula stages. Distinct from gene expression patterns in oocytes and early embryos, the dominant transcripts in sperm represented different functions, including the regulation of motivation (*actb2*) and immunity (*cd74a*, *mhc2dga*).

### 3.3. Gene Co-Expression Modules and Corresponding Biological Functions

To gain insight into the gene regulatory networks that drive early embryogenesis in *C. semilaevis*, we next defined gene co-expression modules along the time series. Among all the expressed genes, we identified 9549 genes that were dynamically expressed in development—defined as genes with a square deviation of VST read count higher than the median across all samples and a TPM of one in at least one sample. Hierarchical clustering of all samples based on these dynamic genes presented a clustering pattern identical to that generated by PCA for all genes (Figure 2A and Figure 3A). Then, we divided the dynamically expressed genes into 33 small co-expression modules by WGCNA [35]. After excluding two modules that resulted from batch effects, we manually merged the remaining modules with similar expression dynamics. This resulted in five representative co-expression modules that accounted for 9448 (98.9%) of all of the dynamically expressed genes. Genes in these five modules exhibited their maximum expression levels in different sample stages. We named these modules Sperm, Maternal, Pre-ZGA, Post-ZGA, and Sperm–post-ZGA, respectively (Figure 3B,C).

Next, we calculated each gene’s intra-modular connectivity and defined hub genes as those with the top 30 connectivities. Gene co-expression networks of hub genes in different modules are shown in Figure 4. The Pre-ZGA module, as the smallest module, presented a weaker intra-modular connectivity than the other four modules.

The GO enrichment analysis on these modules suggested diverse biological functions specific to different growth or developmental tasks (Figure 5; Appendix A). For example, genes highly expressed in the Sperm module mainly conferred immune response and cell activation; genes from the Maternal module—whose transcripts were relatively abundant in oocytes and four-cell to one thousand-cell embryos—mainly contributed to mitosis, transcription inhibition, oogenesis, and sperm–egg fusion; genes from the Post-ZGA module (abundant in the transition from blastula to gastrula) were enriched for cell differentiation, organ development, transcription regulation, and several biosynthetic processes. In addition, genes from the Sperm–post-ZGA module contributed to protein synthesis or disassembly and cellular respiration. Nevertheless, the Pre-ZGA module—which might be composed of relatively slowly degraded maternal transcripts—was not significantly enriched for any GO terms.

### 3.4. Comparative Analysis of Co-Expression Genes between Tongue Sole and Zebrafish

After exploring the gene regulatory networks governing early embryogenesis in *C. semilaevis*, we next questioned whether these networks may also drive embryonic development in other teleost fish. Therefore, we compared the *C. semilaevis* gene expression data with that of *D. rerio* [41] to measure the interspecific conservation based on 14,087 orthologs identified between *C. semilaevis* and *D. rerio*. We first checked for the distribution of orthologs in different co-expression modules. We observed that the orthologs are enriched in the Maternal, Post-ZGA, and Sperm–post-ZGA modules of *C. semilaevis* (Figure 6A).

We next examined the expression dynamics of these orthologs in zebrafish during embryogenesis (Figure 6B). Specifically, the zebrafish orthologs from the Sperm module of *C. semilaevis* showed a very stable expression level throughout embryonic development; orthologs from the Maternal module displayed a higher expression level in early embryogenesis, i.e., from zygote to the end of blastula (0.66–4.66 h post-fertilization, or, hpf), than in mid and later embryonic development; orthologs from the Pre-ZGA module showed the highest expression level from 64-cell (2 hpf) to sphere (4 hpf); orthologs from the Post-ZGA and Sperm–post-ZGA modules remained unchanged from zygote to early segmentation, but slightly rose in the mid and late segmentation stages. Considering that ZGA of zebrafish embryos occurs in the mid-blastula stage [4,43], these results imply that the gene regulatory networks underlying pre-ZGA cleavages are mostly conserved in teleosts, while the networks driving post-ZGA development have been re-wired after the divergence of these two species.

## 4. Discussion

This study generated high-quality RNA-seq data for *C. semilaevis* gametes (sperm and oocyte) and early embryos at different developmental stages (from the four-cell to gastrula stages) to track the global gene expression changes during early embryogenesis. For the first time, we confirmed that large-scale ZGA occurs at the blastula stage in *C. semilaevis*, consistent with the findings for Japanese flounders [44], zebrafish, and frogs [4]. Accordingly, gene transcripts that were abundant in *C. semilaevis* oocytes were also abundant in four-cell to one thousand-cell embryos, which conferred functions that are vital for pre-ZGA cleavage (e.g., the enrichment of mitosis-related genes in the Maternal module). Maternal transcripts may gradually degrade during early embryogenesis, among which the slower ones occupied an increasing proportion of transcripts and presented higher relative abundance before large-scale ZGA. Yet, no significant biological functions were enriched in these slowly degraded transcripts. After large-scale ZGA, the newly generated zygotic transcripts replaced the dominant role of maternal transcripts, leading embryonic development by promoting organ differentiation and the biosynthesis process. Indeed, some genes identified in the Post-ZGA module have been previously verified to play a role in embryonic developmental regulation in *C. semilaevis*, such as *rspo2* (involved in muscle development during embryogenesis) [45,46], *gata5* (increased at the blastula stage and peaked at the heart-beating period) [47], and *sox10* (starting to be expressed in blastula and highly expressed in neurula) [48].

An interspecific comparison revealed similar embryonic regulatory patterns which have been reported in previous gene expression analyses in multiple teleost species; the functional enrichment of cell division or DNA repair has been discovered in two-cell embryos of Atlantic halibut (where cyclin B1 is abundant) [49], zygotes of zebrafish at 3.5 hpf [50], and the morula of both European seabass [51] and three-spined sticklebacks [52]. All these studies observe signals for organ development among genes highly expressed in the later stages—in line with our findings. The consistency of ZGA timing and embryonic transcriptome functions in different teleost species suggests a homology of early embryogenesis in fish. Notably, our transcriptome comparisons between *C. semilaevis* and zebrafish revealed conserved gene regulatory networks underlying pre-ZGA cleavages but not post-ZGA development in fish (Figure 6). The interspecific conservation and divergence of gene expression is also noteworthy for aquaculture researchers, as it may necessitate a precise adjustment in the target growth stage during the development of breeding techniques across distinct species.

Interestingly, genes highly expressed in the sperm of *C. semilaevis* conferred actin- and immune-related functions. The abundance of actin was expected, as actin is crucial for spermatogenesis in animals as it forms the filaments that support capacitation and motility [53,54]. However, the strong correlation between sperm transcripts and the immune system in fish has rarely been reported. Previous studies have primarily limited the functions of the sperm transcriptome to sperm capacity, fertilization, and embryonic development [55,56,57]. Immune regulatory functions in mammalian sperm and semen transcripts are assumed to protect the maternal genital tract from infection or inhibit maternal immune responses to sperm [56,57,58]. Transcriptome analysis of the ovaries of rockfish—a viviparous teleost fish that stores sperm inside the ovary cavity for months—have likewise revealed changes in immune function before and after mating [59]. Together, these studies emphasize the effect of semen on maternal immune regulation during in vivo fertilization, while the immunological effect of semen among in vitro fertilized species remains unclear. Our functional analysis of sperm-specific transcripts in *C. semilaevis* in this study suggests that sperm cells may promote immune activation and, therefore, be critical for safeguarding in vitro fertilization in fish, which is worthy of further research.

In conclusion, our study comprehensively analyzes gametes and early embryonic gene expression profiles in *C. semilaevis*. The high-depth RNA-seq dataset will serve as a valuable resource for studying gene regulatory mechanisms underlying the early life of Cynoglossidae. It may assist in searching for candidate genes (e.g., related to growth and development) that benefit aquaculture practices. The transcriptomic comparison between species also contributes to knowledge about the conservation and divergence of gene regulation in fish early embryogenesis. Future studies are expected to explore the evolutionary implications of these regulatory strategies for speciation and adaptation.

## Figures and Tables

**Figure 1 life-14-00505-f001:**
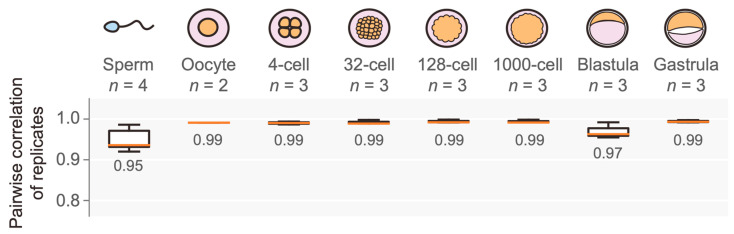
A schematic for sample collection with the number of replicates indicated (**top**) and the high consistency of gene expression between replicates (**bottom**). The degree of consistency was measured by Pearson’s correlation of gene expression between two replicates within the same sample group. The mean correlation coefficient for each sample group is presented under the box.

**Figure 2 life-14-00505-f002:**
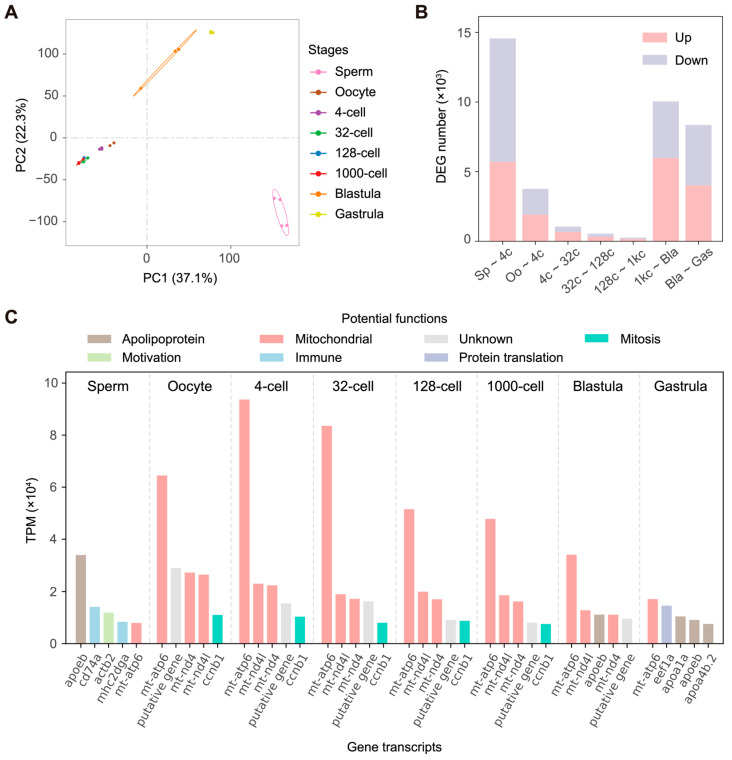
(**A**) Principal component analysis (PCA) based on 21,920 *Cynoglossus semilaevis* genes covered by at least one read. (**B**) Numbers of up-regulated (pink) and down-regulated (grey) differentially expressed genes (DEGs) between pairs of consecutive embryogenesis stages in *C. semilaevis*. Sp: sperm; Oo: oocyte; 4c: 4 cells; 32c: 32 cells; 128c: 128 cells; 1kc: 1000 cells; Bla: blastula; Gas: gastrula. (**C**) The five most abundant gene transcripts in different gametes and embryonic developmental stages in *C. semilaevis*. The potential functions of genes are indicated by color. The putative gene (gene ID: Cse_R021186, located in scaffold7208: 12,125–16,516) was structurally predicted but did not correspond to orthologs in the current NCBI-nr database; its biological function is unclear. TPM, transcripts per million.

**Figure 3 life-14-00505-f003:**
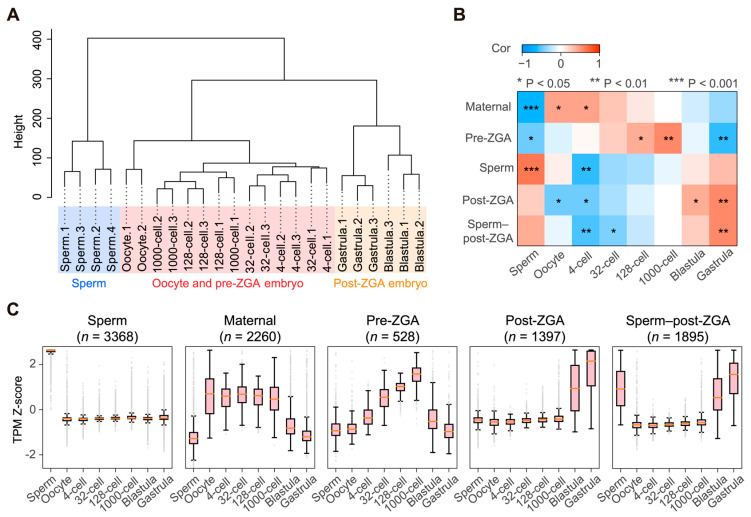
(**A**) Hierarchical clustering of the 24 samples based on 9549 dynamically expressed genes. Three distinct clusters of samples are highlighted by different colors and named after the developmental stages, namely “Sperm” (blue), “Oocyte and pre-ZGA embryo” (red), and “Post-ZGA embryo” (yellow). (**B**) The correlation (Cor) between the eight sample categories and five co-expression modules. The significance (P) of Spearman’s correlation analysis is indicated by asterisks. (**C**) Relative expression levels of genes in different modules, represented by Z-score of transcripts per million (TPM). The title of each plot indicates the highly expressed stage(s) of gene members in the relevant module. The number of genes is shown in parentheses. Large-scale zygotic genome activation (ZGA) was inferred to occur in the blastula stage.

**Figure 4 life-14-00505-f004:**
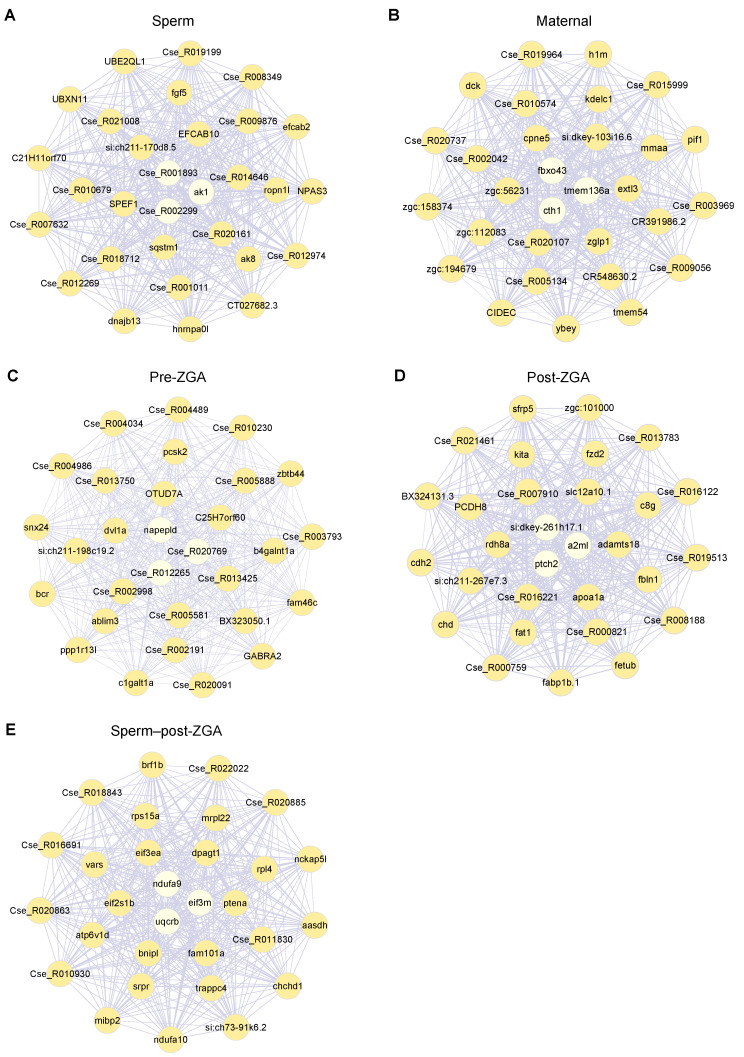
Hub genes with the top 30 intra-modular connectivity in five co-expression modules, respectively, namely Sperm (**A**), Maternal (**B**), Pre-ZGA (**C**), Post-ZGA (**D**), and Sperm–post-ZGA (**E**). Purple lines represent the correlation (0.95~1) between two linked genes, with line width representing value size. Genes closer to the center of the network (indicated by light yellow) have higher intra-modular connectivity.

**Figure 5 life-14-00505-f005:**
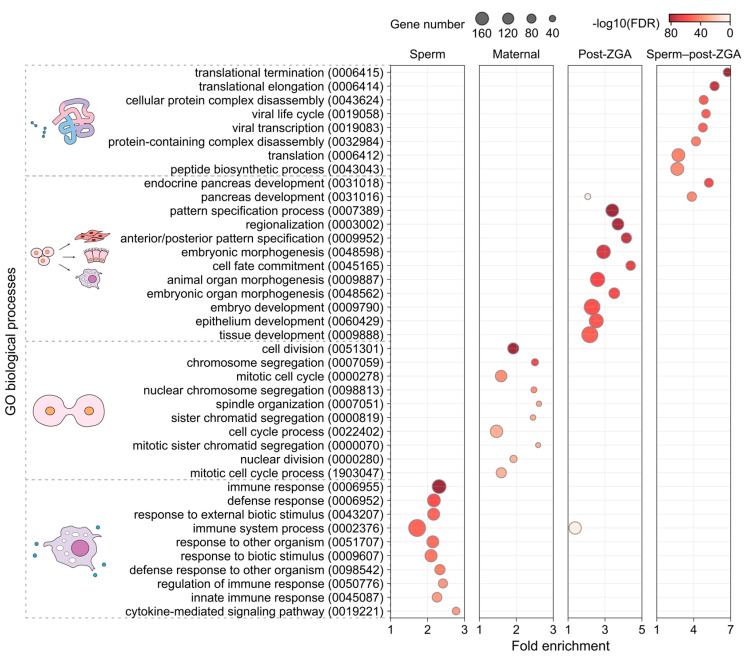
The top most significant Gene Ontology (GO) biological processes which are enriched in different gene co-expression modules, respectively. GO ID of each term is indicated in parentheses. Biological processes with similar functions are clustered and represented by a single icon. FDR: false discovery rate.

**Figure 6 life-14-00505-f006:**
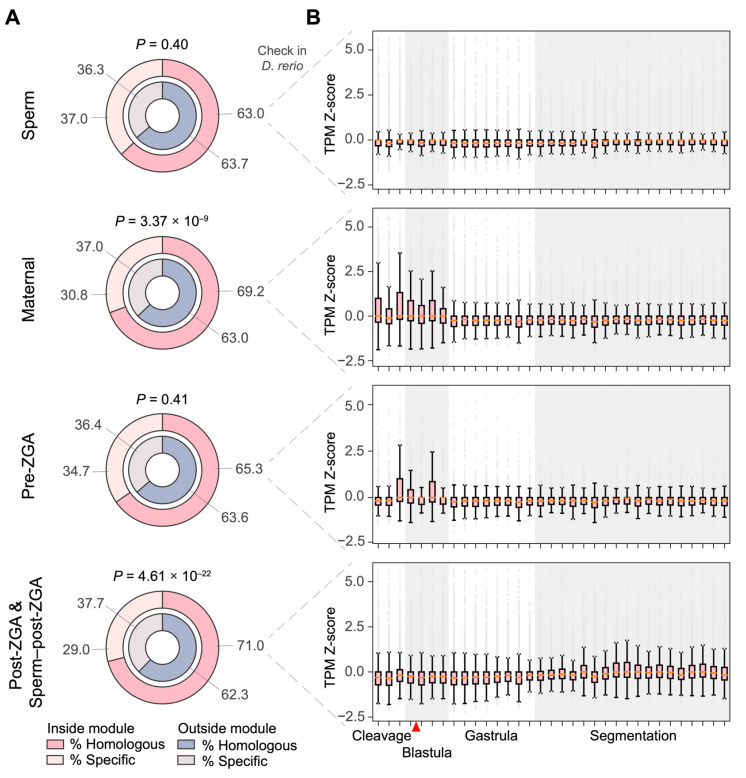
(**A**) Interspecific conservation of genes in co-expression modules, compared with *Danio rerio*. The distribution bias of orthologs was tested by Fisher’s exact test. (**B**) Relative expression level of the orthologs in *D. rerio* during embryogenesis. The horizontal scale indicates samples taken every 40 min from fertilization. The red arrow indicates the putative ZGA of the *D. rerio* embryo. The *D. rerio* gene expression data were obtained from Levin et al. [41].

## Data Availability

The raw RNA-seq data supporting this study’s findings have been deposited into the CNGB Sequence Archive (CNSA) (https://db.cngb.org/) with accession number CNP0001602, accessed on 10 April 2024.

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
