# Peer review of "Unveiling Gene Expression Dynamics during Early Embryogenesis in Cynoglossus semilaevis: A Transcriptomic Perspective"

_life, 2024, doi:10.3390/life14040505_

Round 1

Reviewer 1 Report (Previous Reviewer 1)

Comments and Suggestions for Authors

In this manuscript, the authors analyzed the transcriptomic profiling on two types of gametes (oocytes and sperms) and early embryos (ranging from the four-cell to the gastrula stage) of an economically valuable flatfish – Chinese tongue sole Cynoglossus semilaevis. They found that large-scale zygotic genome activation (ZGA) occurs in the blastula stage, and demonstrated distinct functional enrichments across different gametes/developmental stages. This work is interesting. Generally, the experiments were well designed and presented. I recommend it for publication in this journal.

Author Response

Dear reviewer,

We are particularly grateful for your positive remarks on our work. Your recognition of the strengths within the manuscript serves as a significant encouragement to our team.

Thank you for the time and effort you dedicated to reviewing our manuscript.

Best regards,

Xinyi Cheng

Reviewer 2 Report (Previous Reviewer 2)

Comments and Suggestions for Authors

Notes to the authors of the manuscript:

The manuscript was revised mainly according to the comments and recommendations of the reviewers. The revised version meets the requirements of the scientific journal to publish high-quality articles that are useful both for the scientific community and for the relevant aquaculture industry. This increases the scientific-applied value of the manuscript and increases the reader's interest and citation potential of the article. I recommend the authors to continue research in this scientific field with other teleosts to investigate the conservation and diversification of early embryogenesis in fish.

Author Response

Dear reviewer,

We would like to express our appreciation for the time and expertise you dedicated to reviewing our revised manuscript. We are delighted that our revisions were met with approval.

Our team will further our research in the inter-species conservation of embryogenesis in teleost fish. We believe that more valuable scientific phenomena can be revealed in the future.

Thank you for your continued support.

Warm regards,

Xinyi Cheng

This manuscript is a resubmission of an earlier submission. The following is a list of the peer review reports and author responses from that submission.

Round 1

Reviewer 1 Report

Comments and Suggestions for Authors

In this manuscript, the authors analyzed the transcriptomic profiling on two types of gametes (oocytes and sperms) and early embryos (ranging from the four-cell to the gastrula stage) of an economically valuable flatfish – Chinese tongue sole Cynoglossus semilaevis. They found that large-scale zygotic genome activation (ZGA) occurs in the blastula stage, and demonstrated distinct functional enrichments across different gametes/developmental stages. This work is interesting. Generally, the experiments were well designed and presented. I have a few concerns that are listed below. 

Major concerns

1. In Materials and Methods, the authors prepared and sequenced the RNA-seq libraries by using two different methods. Why? Are the sequencing results obtained from these two methods comparable?

2. As shown in Figure 1, only two replicates of oocytes were used in this study. Why? 

Minor concerns

1. Line 102-103, should “between 20 and 20 kb” be “between 20 bp and 20 kb”?

2. Line 179-180, except at different stages of embryonic development in Csemilaevis, the authors also analyzed the five most abundant transcripts for genes in sperm and oocyte.

3. Line 242, Should “contribute to” be “contributed to”?

Comments on the Quality of English Language

In this manuscript, the authors analyzed the transcriptomic profiling on two types of gametes (oocytes and sperms) and early embryos (ranging from the four-cell to the gastrula stage) of an economically valuable flatfish – Chinese tongue sole Cynoglossus semilaevis. They found that large-scale zygotic genome activation (ZGA) occurs in the blastula stage, and demonstrated distinct functional enrichments across different gametes/developmental stages. This work is interesting. Generally, the experiments were well designed and presented. I have a few concerns that are listed below. 

Major concerns

1. In Materials and Methods, the authors prepared and sequenced the RNA-seq libraries by using two different methods. Why? Are the sequencing results obtained from these two methods comparable?

2. As shown in Figure 1, only two replicates of oocytes were used in this study. Why? 

Minor concerns

1. Line 102-103, should “between 20 and 20 kb” be “between 20 bp and 20 kb”?

2. Line 179-180, except at different stages of embryonic development in Csemilaevis, the authors also analyzed the five most abundant transcripts for genes in sperm and oocyte.

3. Line 242, Should “contribute to” be “contributed to”?

Reviewer 2 Report

Comments and Suggestions for Authors

Positive notes to the authors:

1. The subject of the manuscript is particularly relevant, as it is related to revealing the dynamics of gene expression during early embryogenesis in Cynoglossus semilaevis from a transcriptomic perspective;

2. The authors have done a detailed literature review and have shown the state of the art in the field of biochemical ichthyogenetics. Nevertheless, the dynamics of gene expression governing early embryogenesis of Cynoglossus semilaevis remains unclear in most metazoan lineages, and this remains a scientific task of research;

3. In the Material and methods section, sampling, RNA extraction and RNA sequencing, as well as gene expression analysis are described in detail. In addition, a homology analysis was made between the embryonic transcriptome of tongue sole and zebrafish;

4. The results are relatively well visualized using 6 figures, which could be 10-15% larger scale for better visualization;

5. The erudition of the authors is evident in the quality of the discussion in each scientific article. Here the authors have tried to dig deep into the bowels of biochemical ichthyogenetics. However, they could look in more detail at the expression of genes in sole, as well as regulatory mechanisms in the early life of Cynoglossidae;

6. A cursory attempt has been made to summarize the results and outline future research, but there are no specific recommendations to the fish farming industry.

Negative notes and recommendations to the authors:

1. At the end of the introduction, the specific purpose of the scientific research, which corresponds to the title of the manuscript, is not indicated;

2. Some figures especially 4 and 5 need to be scaled up by 10-15% for better visualization;

3. In order for the research to be scientific and applied, it is necessary for the authors to formulate useful recommendations for fish farming practice, especially since the object of the research is an economically valuable type of fish.

Comments on the Quality of English Language

Evaluation of the English language of the manuscript:

The article is written in fairly professional English and does not need serious corrections. However, I recommend a final polish by an English-speaking editor.